# A new approach to the temporal significance of house orientations in European Early Neolithic settlements

Nils Müller-Scheeßel[1] *, Johannes Müller[1], Ivan Cheben[2], Wiebke Mainusch[1], Knut Rassmann[3], Wolfgang Rabbel[4], Erica Corradini[4], Martin Furholt[5]

**1** Institute for Pre- and Protohistoric Archaeology, Christian-Albrechts-University, Kiel, Germany, **2** Archaeological Institute of Slovak Academy of Sciences, Nitra, Slovakia, **3** Romano-Germanic Commission, German Archaeological Institute, Frankfurt a. M., Germany, **4** Institute of Geosciences, Christian-Albrechts-University, Kiel, Germany, **5** Department of Archaeology, Conservation and History, University of Oslo, Oslo, Norway

* nils.mueller-scheessel@ufg.uni-kiel.de

**Data Availability Statement:** The relevant data is available from DANS EASY: https://doi.org/10.17026/dans-xu6-c6yr.

## Abstract

This paper shows that local differences in house orientation in settlements from the Early Neolithic in Central Europe reflect a regular chronological trajectory based on Bayesian calibration of $^{14}$C-series. This can be used to extrapolate the dating of large-scale settlement plans derived from, among other methods, geophysical surveys. In the southwest Slovakian settlement of Vráble, we observed a progressive counter-clockwise rotation in house orientation from roughly 32˚ to 4˚ over a 300 year period. A survey of published and dated village plans from other LBK regions confirms that this counter-clockwise rotation per settlement is a wider Central European trend. We explain this observation as an unintentional, unconscious but systematic leftward deviation in the house builders' cardinal orientation, which has been termed "pseudoneglect" in studies of human perception. This means that whenever houses were intended to be oriented towards a specific direction and be parallel to each other, there was an error in perception causing slight counter-clockwise rotation. This observation is used as a basis to reconstruct dynamics of Early Neolithic settlement in the Slovakian Žitava valley, showing a rapid colonization, followed by increased agglomeration into large villages consisting of strongly autonomous farmsteads.

## Introduction

The onset of large-scale magnetic surveys during the last decade has offered completely new possibilities for the study of settlement systems. Within European archaeology, among others, this is especially relevant for the reconstruction of Chalcolithic mega-settlements of eastern Europe [1], Neolithic enclosures all over Europe [2], and also settlements of the earliest Neolithic group in Central Europe, the so-called *Linearbandkeramik* (LBK) (e. g., [3, 4]).

Using excavation results and Bayesian dating and combining this chronological information with magnetic plans, we propose that it is possible to make substantial claims about the

**Funding:** This research was funded by the Deutsche Forschungsgemeinschaft (DFG, German Research Foundation; project no. 2901391021 – SFB 1266; PIs: J. M., W. R., M. F.) and the Vedecká grantová agentúra MŠVVaŠ SR a SAV (VEGA; project no. 2/0107/17; PI: I. Ch.). DFG: https://www.dfg.de/en/index.jsp VEGA: https://www.minedu.sk/vedecka-grantova-agentura-msvvas-sr-a-sav-vega/ The funders had no role in study design, data collection and analysis, decision to publish, or preparation of the manuscript.

**Competing interests:** The authors have declared that no competing interests exist.

settlement history of a whole region without the need to excavate large areas. As Bayesian [14]C-calibration shows that the orientation of houses follows a chronological order, orientation can be used as a proxy for the chronological order of houses. We see the change in orientation as the concrete outcome of a neurobiological phenomenon, pseudoneglect, which hitherto has been attested only in laboratory studies. The pseudoneglect hypothesis is relevant where there are no other stronger constrains influencing building activities (topography, circular arrangement of houses, proximity of houses, but also social factors like overarching authorities, etc.).

Our study takes the so-called *Linearbandkeramik* as example, which is inextricably linked with the onset of Neolithic lifeways in Central Europe at around 5300 cal BCE. What sets LBK settlements apart from any other archaeological culture, is their use of very characteristic long houses. These are massive buildings of lengths of 30 meters and more, the roof supported by solid posts of oak, always set in cross-rows of three. Additionally, the houses are uniquely marked by long clay extraction and refuse pits at each side. These characteristics make LBK settlements and houses easily identifiable within magnetic surveys and an ideal test-case for our approach.

Especially huge rescue excavations like that on the Aldenhovener Platte have pioneered large-scale settlement studies in terms of settlement systems and networks of the LBK. They are, however, mostly still based on typochronological instead of scientific dating methods [5, 6], as the excavations were already done before the main [14]C-boom in archaeology. In other cases, larger geophysical surveys combined with small-scale excavations have produced important insights into the general layout of LBK houses and settlements [7, 8]. Nevertheless, without the extremely time-consuming large-scale excavations such as those carried out on the Aldenhovener Platte, the understanding of the diachronic developments of the settlements are necessarily much coarser (e. g., [9] for the Wetterau in Central Germany). Therefore, we propose a new approach to understanding the chronological development of large-scale settlements without extensive excavation: dating by house orientation.

The idea that house orientation holds significance has been around for a long time in LBK research, but the discussion of regional differences in orientation always trumped considerations of variance in house orientation at individual sites (for an overview of approaches to orientation of LBK houses see [10], 529ff.). It is, however, at the local, site-specific level that house orientation can be used for relative chronological dating. Therefore, we are not interested in the differences in orientation of houses between, say, Slovakia and the Alsace, but only in those differences per site or per micro-region.

E. Sangmeister ([11], 91ff.) was the first to point out the variation in house orientation within the same settlement. Taking the settlement of Köln-Lindenthal as an example, he grouped the houses into seven classes of the same orientation (within 2–3˚) and postulated the contemporaneity of houses within these groups. Apart from the similar orientation, the fact that all houses of each group were found across the whole settlement was, in his eyes, a very strong argument. He assumed that the settlements were temporarily abandoned and, when resettled, the houses were built with a slightly different orientation ([11, 12], 454). Despite the fact that he found similar patterns in other settlements, his suggestion was never adopted and later was even refuted on the grounds of overlapping houses with the same orientation ([13], 224f.) or ceramic typology ([14], 106). However, taking magnetic prospections in southwest Slovakia as point of departure, we will show that the findings of Sangmeister should have been taken more seriously.

During our fieldwork in Vráble we conducted a comprehensive dating program and thus produced a dataset that is large enough to compare orientation and dating of houses in detail. A comparison of these two measures proves their correlation and the testing of this dating method at other sites suggests that orientation could play a key-role in deciphering the development of large-scale settlements as referred to above. Under certain conditions orientation could serve as

a proxy for relative chronology, and in addition, when combined with small-scale excavations for $^{14}$C-dating can also be integrated into the absolute chronology of a specific site.

## Materials and methods

The Žitava valley in southwest Slovakia (Fig 1) was settled during the first expansion of the LBK [15], which is now assumed to have taken place after 5350 BC ([16], 138). The subsequent LBK settlement of the Upper Žitava-valley–the main focus of our project–is well documented in regional databases [17–19] and through our intensive fieldwork. Through magnetic prospections covering an area of about 2 km$^2$, the layout of 13 settlements is known (Fig 2). The site of Michal nad Žitavou [20] is not taken into account in this number because the areas measured in 2011 were too small to yield reliable results.

Excavations were carried out in the three settlements of Vráble (Fig 3) uncovering a total of 15 houses. The ceramic material unanimously date these houses to the youngest LBK, which is also supported by the 99 $^{14}$C-dates obtained. A large number of additional $^{14}$C-dates from nine other houses was gained by a targeted coring program [21].

Our excavations in the Žitava-valley have shown (like in other cases, cf. [8]) that the agreement between the excavated features and the magnetic picture is very high [22, 23]). Because LBK houses are unique and easily identified by their characteristic long pits bracketing the long sides of the house, it seems permissible to reconstruct the number and placement of houses on the basis of the magnetic measurement. Therewith, size and orientation of houses can also be reasonably determined. This methodology has successfully also been applied elsewhere on LBK settlements (e. g. [3], 83 Fig 7).

It is likely that the actual number of houses in ancient times was higher than is estimated by this method. In some instances, long pits might have already eroded away, in others they might not be filled by significant amounts of material with high magnetic susceptibility. Finally

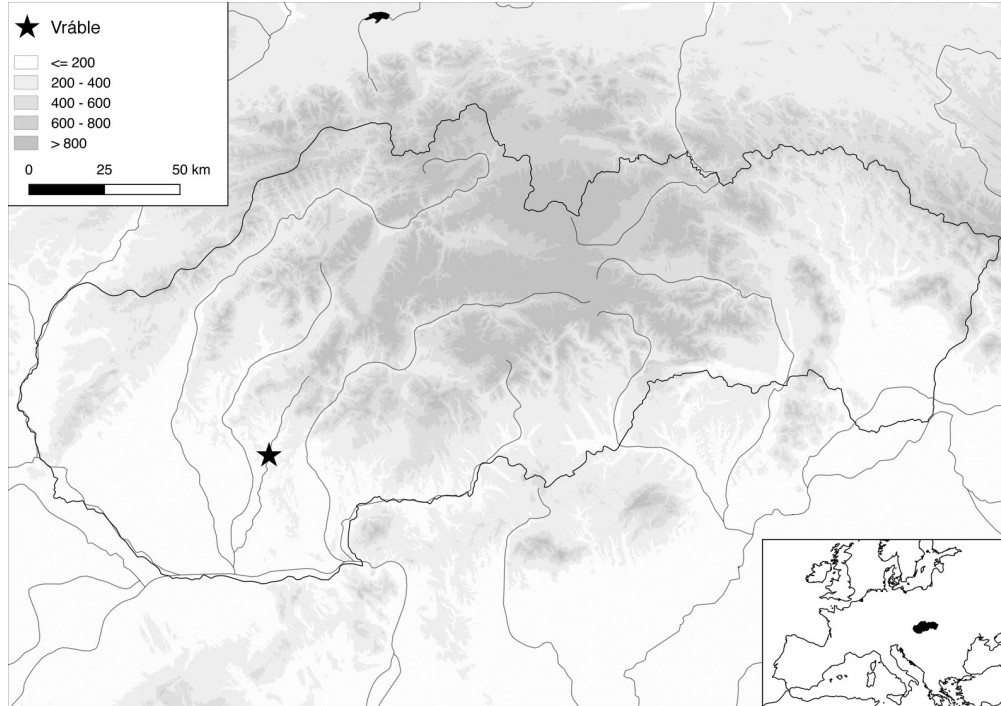

**Fig 1. Terrain map of Slovakia and surrounding region, showing the location of Vráble 'Véľke Lehemby' and the Žitava-valley.**

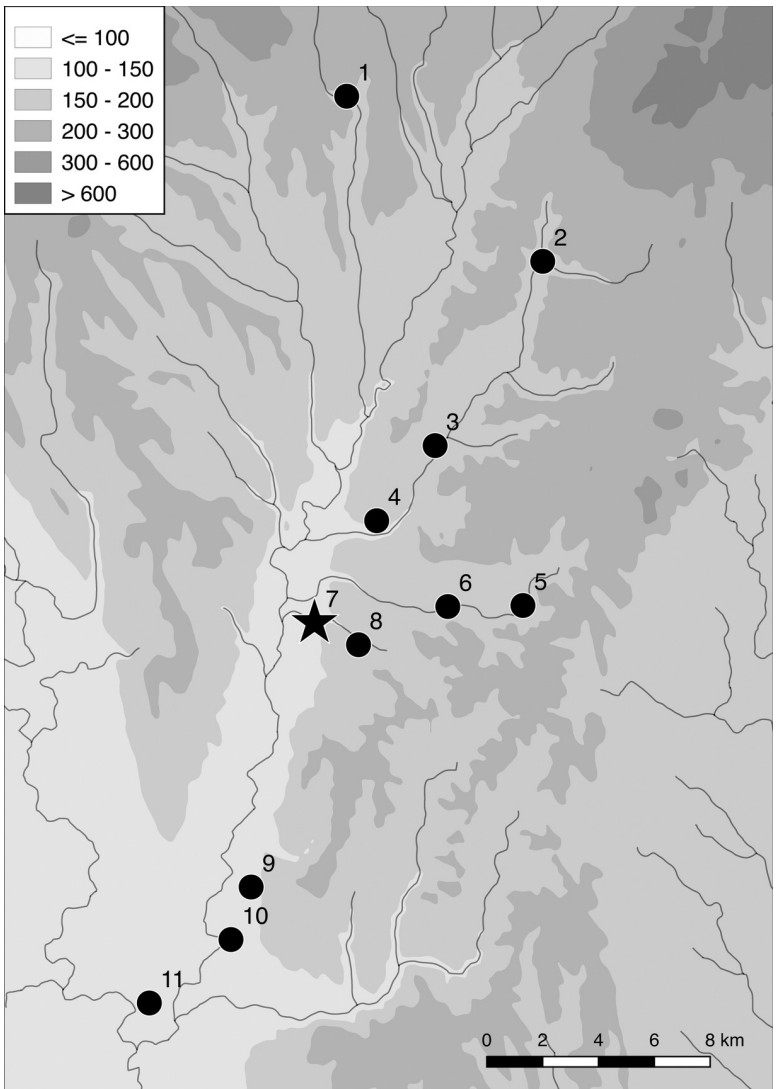

**Fig 2. The Žitava-valley with the LBK sites with magnetic imagery.** 1 Sl̓ažany 'Na Domovine'; 2 Čierne Kl̓ačany 'Mlynské diely'; 3 Nevidzany 'Konopiská'; 4 Horný Ohaj 'Dolne siatie'; 5 Čifare 'Kapustniská'; 6 Telince 'Horné lúky'; 7 Vráble 'Vel̓ke Lehemby'; 8 Vráble 'Drakovo'; 9 Maňa 'Za hlbokou cestou'; 10 Vlkas 'Do hulského chotára'; 11 Úl̓any nad Zitavou 'Dolné diely'.

there might be such a large degree of overlap between adjacent house pits that they are no longer distinguishable in the magnetic picture. All of these factors lead to an underestimation of house numbers, never to an overestimation. For the present context it is especially important that the orientation of houses is not be affected by these possible taphonomic processes.

Currently it is not possible to reliably identify archaeological features in magnetic plans automatically (for a semi-automatic approach see [24]). This is due to the very low magnitude of signal of the features on the one hand, and the still greater variability of the signal on the other. Automatic classification thus leads either to the marking of only the most distinctive features or virtually the whole area, depending on the chosen sensitivity.

Therefore, it seemed more fruitful to perform an expert identification of features focusing first on delimiting the long pits and successively on the houses. For that purpose a polygon was drawn between two long pits seemingly belonging together based on orientation, position and

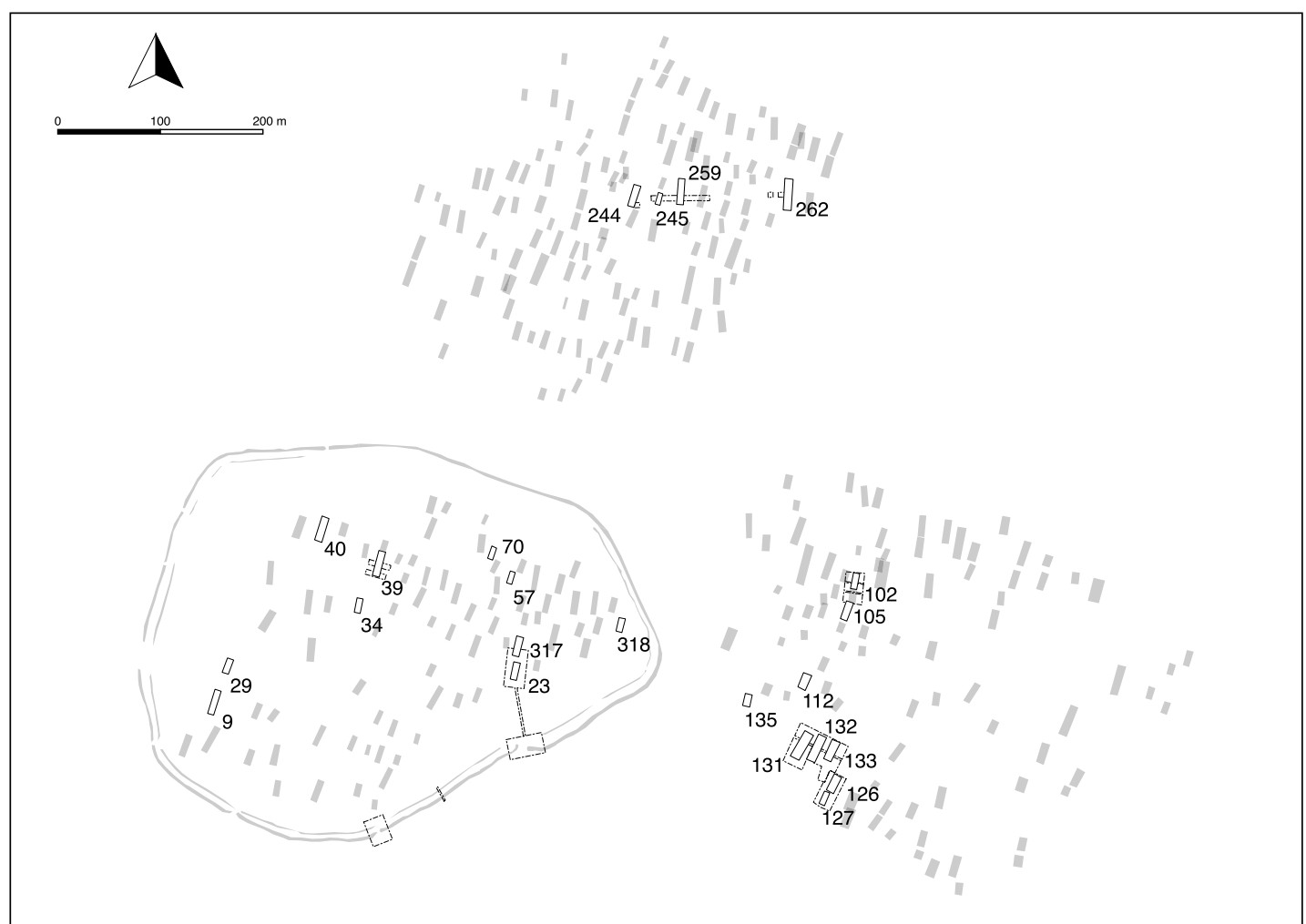

**Fig 3. The site plan of Vráble 'Véĺke Lehemby' derived from magnetic images with three distinct LBK settlements with excavation trenches and dated houses.**

length. This was done for each site by three of the authors (M. F., W. M., N. M.-S.) independently. Houses were only accepted which were identified by at least two of the authors, i.e. where the area of overlap between polygons drawn by different researchers was equal to at least half of the area of the largest polygon.

In order to avoid human measurement errors, the orientation of individual houses was calculated automatically. The orientation was determined by taking the average of the orientation of the long sides of each house as expressed by the reconstructed polygons (minimum number of long sides = 4 as at least two polygons identified by two experts per house). In this way, it is also possible to estimate the standard deviation of orientation for each house.

As it is normally not possible to identify an unequivocal single direction for houses, we reported orientations by their degree closest to true north. Thus, orientations range from 270˚ (west–east) over 360˚/0˚ (north–south) to 89.9˚ (eastnortheast–westsouthwest). According to this system, values between 90˚ and 269.9˚ cannot occur.

In previous studies, orientation data is usually aggregated by deviation from north (e.g. +10, -15) and by treating it like linear data. However, for directional data like the orientation of houses standard measures as used for linear data (like arithmetic mean) is inappropriate ([25], 12). The calculation of average degree is very straightforward, though, when angles are

**Table 1. The LBK-sites in the Žitava-valley with descriptive statistics on sizes and number of houses.**

| Site | | Magnetic survey in ha | Supposed size of settlement | Area of settlement covered | Percentage of settlement covered | Number of houses (at least two experts agreed) | Percentage of agreement between all three experts | mean orientation in° | standard deviation in° | r | Reference |
|---|---|---|---|---|---|---|---|---|---|---|---|
| Čierne Kľačany 'Pri mlyne' | | 11.5 | unknown | 7.4 | - | 49 | 63.3 | 23.4 | 5.21 | 0.996 | Cheben 2015, 114 Fig 3A |
| Čifare 'Kapustniská' | | 15.7 | 8.8 | 8.1 | 92% | 41 | 41.5 | 16.1 | 5.64 | 0.995 | unpubl. |
| Horný Ohaj 'Dolne siatie' | | 1.6 | unknown | 0.8 | - | 11 | 45.5 | 12.2 | 4.73 | 0.997 | unpubl. |
| Maňa 'Za hlbokou cestou' | | 27.9 | 5.9 (3.1 + 1.3 + 1.5) | 5.7 | 97% | 29 | 34.5 | 24.5 | 7.33 | 0.992 | unpubl. |
| Nevidzany 'Konopiská' | | 5.5 | unknown | 2.7 | - | 22 | 40.9 | 19.8 | 6.96 | 0.993 | unpubl. |
| Sľažany 'Domovina' | | 0.84 | unknown | 0.84 | – | 6 | 33.3 | 18.9 | 6.87 | 0.993 | Cheben 2015, 113 Fig 2 |
| Telince 'Horné lúky' | | 8.0 | 4.2 | 4.1 | 98% | 13 | 38.5 | 8.2 | 7.93 | 0.990 | unpubl. |
| Úľany nad Zitavou 'Dolné diely' | | 8.2 | 4.4 | 3.0 | 68% | 34 | 61.8 | 14.7 | 11.38 | 0.980 | unpubl. |
| Vlkas 'Do hulského chotára' | | 17.7 | 8.7 (8.0 + 0.7) | 8.5 | 98% | 61 | 42.6 | 24.4 | 5.24 | 0.996 | unpubl. |
| Vráble 'Drakovo' | | 1.4 | unknown | 0.3 | - | 2 | 50.0 | 25.8 | 7.87 | 0.991 | unpubl. |
| Vráble 'Veľke Lehemby' | north | > 1 sqkm | 11.9 | 11.5 | 97% | 124 | 63.7 | 15.2 | 6.11 | 0.994 | Furholt et al. 2014, 230 Fig 3 |
| | southeast | | 14.0 | 14.0 | 100% | 89 | 49.4 | 18.0 | 9.15 | 0.987 | Furholt et al. 2014, 230 Fig 3 |
| | southwest | | 8.6 (enclosure: 14.5) | 8.6 | 100% | 100 | 38.0 | 16.3 | 7.76 | 0.991 | Furholt et al. 2014, 230 Fig 3 |

transformed to vectors ([25]). All aggregated orientation data (average, standard deviation) reported in this paper therefore derives from calculations of vectors, except where otherwise noted (in cases of already tabulated data from other studies).

To understand the internal organizations of the settlements, we computed the distances between the Next Neighbors and also between all houses. For reasons of simplicity, we chose the centroid of each house as spatial representative.

## [14]C-dates

From the 2014 excavations datable material was available for five houses of the northern settlement (for an in-depth-discussion of the [14]C-dates from Vráble see [21]). In the southeastern settlement, seven houses yielded datable material during excavations in 2013 and 2016. Two further houses in the southeastern settlement were dated by charcoal samples from test cores. Because of a targeted coring program, the largest number of houses with the most even spatial distribution derives from the southwestern settlement. There, it was possible to extract suitable, short-lived material from cores for seven houses. In addition, three houses were investigated by excavation in the years 2012 and 2017. Of the 100 houses of the southwestern settlement, ten, i.e. 10%, are therefore datable. Considering all three settlements together, there is datable material from 24 houses. In relation to the 313 houses defined by the experts in the magnetic plan of this settlement cluster, this equals 8%.

It is a common opinion that the material in the long pits derives from the time after the initial construction of LBK houses. If they were filled during the use-life of the house or after the house's abandonment is a matter of debate [26–28]). Nevertheless, it should be obvious that simply taking the oldest date from each long pit would not be appropriate if we want to link the orientation of each house to the moment of its erection, especially as the number of [14]C-dates per house varies. Therefore, we decided to rely on the function „First"in the calibration tool OxCal v. 4.2.4 [29] to generate simulated oldest dates for each house when two or more plausible dates are available. This was possible for 17 of the 24 dated houses.

## Results

### Identification of houses

As the result of the expert identification, a total of 581 anomalies in 13 sites were identified as LBK longhouses (Table 1). There are differences between the sites: While ground plans identified by only two of the three experts predominate at most sites, the expert identifications are more unanimous at some settlements. This applies in particular to the sites Čierne Kľačany and Úľany nad Zitavou. Vráble is also dominated by houses where all three experts agree. The degree of agreement is likely an indication of the state of preservation and thus the clarity of the magnetic anomalies, or could possibly also reflect building density as with overlapping houses the chance that the same long pit is assigned to different houses by the experts rises.

### Orientation of houses

Within the Upper Žitava Valley the orientation of the houses is relatively uniform. The average orientation is 18.0˚ with 0.0407 circular variance. Nevertheless, the range of measured degrees covers more than one eighth of a full circle (> 45˚) and includes orientations between 355.6˚ and 44.2˚. If the data are broken down to settlement level, there are also clear differences. The mean values for each settlement range between 8.2˚ and 25.8˚. Within the settlement cluster of Vráble the differences are very limited: the average orientations are 15.2˚, 16.3˚ and 18.0˚ for the northern, southwestern, and southeastern settlement respectively.

The average distance between house centroids of the settlements ranges between 16.0 and 40.0 m, signaling differences in building densities. In relation to the difference in orientation between neighboring houses, an interesting negative correlation surfaces (Fig 4): It seems that with higher distances between nearest neighbors, the differences in average orientation decrease. Likewise, higher density equals stronger differences in orientation.This means that with a larger distance between houses (equaling a lower density of the settlement layout), the variability of house orientation decreases.

Furthermore, we see that houses with similar orientations are spaced at regular intervals (Fig 5). The maxima of kernel density estimates for the distances of houses showing a difference in orientation of 4˚ or less peak at distances of around 75 and 150 m, distances which are much higher than that to the nearest neighbor.

### Dating of houses at Vráble

In the Bayesian model of [14]C dates from Vráble presented by [21], the occupation of the settlement cluster is dated to ca. 5250–4950 BCE and the duration of house use is on average 40 years. House orientation was not part of the modeling process. Therefore, it is far from self-evident that we find a correlation between the modeled first dates (represented by their weighted mean) and the orientation of the respective houses (Fig 6). For the three settlement parts treated as a whole, the correlation-coefficient r is only moderately strong (0.30). Remarkably,

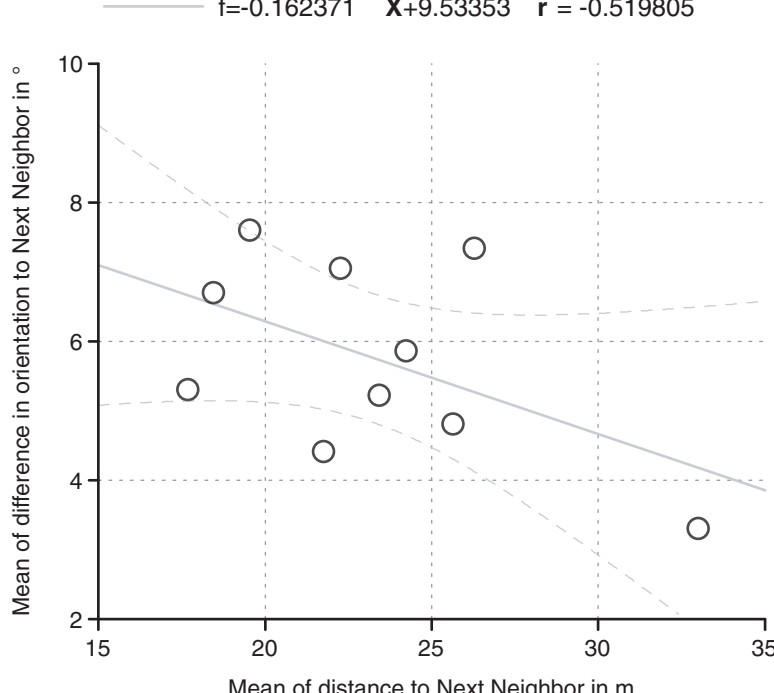

**Fig 4. Regression line with 95%-confidence interval of arithmetic mean of distances and circular mean difference of orientation between nearest neighbors/houses (only settlements with magnetic measurements of areas of 5 +ha).**

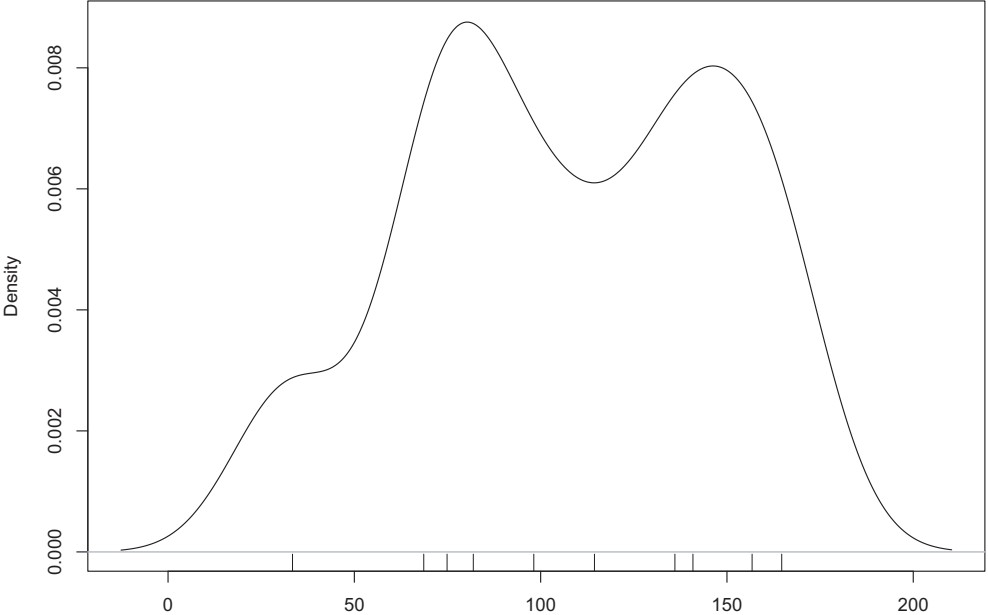

**Fig 5. Kernel density estimation (Gaussian Kernel, bandwidth = 15 m) of the maxima of the distances between houses with similar orientation (difference $< = 4°$) (n = 10; only settlements with magnetic measurements of areas of 5+ha).**

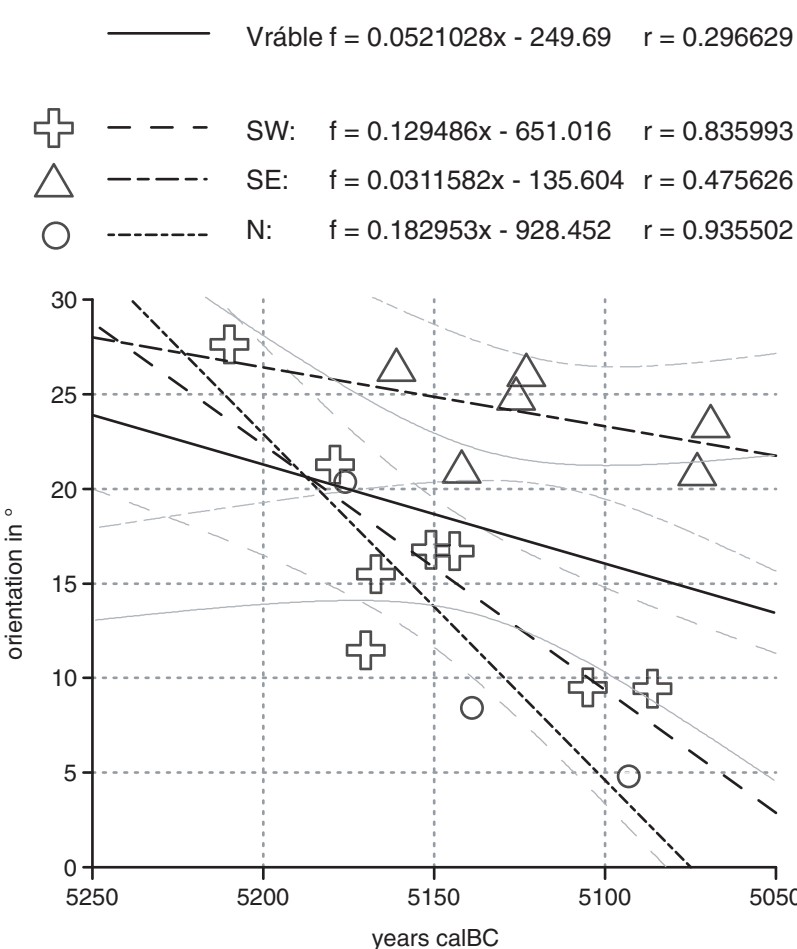

**Fig 6. Relationship between the modeled first dates (represented by their weighted mean) and the orientation of the respective house of Vráble.** Regression line with 95%-confidence intervals for Vráble as a whole and each of its three settlements (see text).

it is much higher when looking at the individual settlement. Generally, the correlations between the first dates and orientation are high for the southwestern and northern settlements (0.94 and 0.84 respectively), while in the southeastern settlement, the correlation is relatively loose (0.48). This can, however, be explained by the fact that in the last settlement an exceptional group of houses was selected for excavation: a group of four houses standing so close together that we initially expected a contemporary house cluster. Through excavation, we could demonstrate that this group of houses actually represents a time-depth of 200 years (5200–5000 BCE). The unusually close proximity of these houses prevented the deviation in house orientation over time that we can observe in the other areas of the settlement.

## Testing the correlation: Diachronic change in house orientation across LBK Europe

A tendency of older houses towards a more northerly direction and of younger houses towards a more north-western orientation–i.e. a leftward shift in orientation–was also observed in the Merzbachtal in western Germany ([30], 925 with fig 741). Despite the fact that E. Mattheußer ([31], 10) argued that there is no general tendency in changing house orientation over time, her own data tells a different story (Fig 7A). While houses of older phases show an average orientation of about

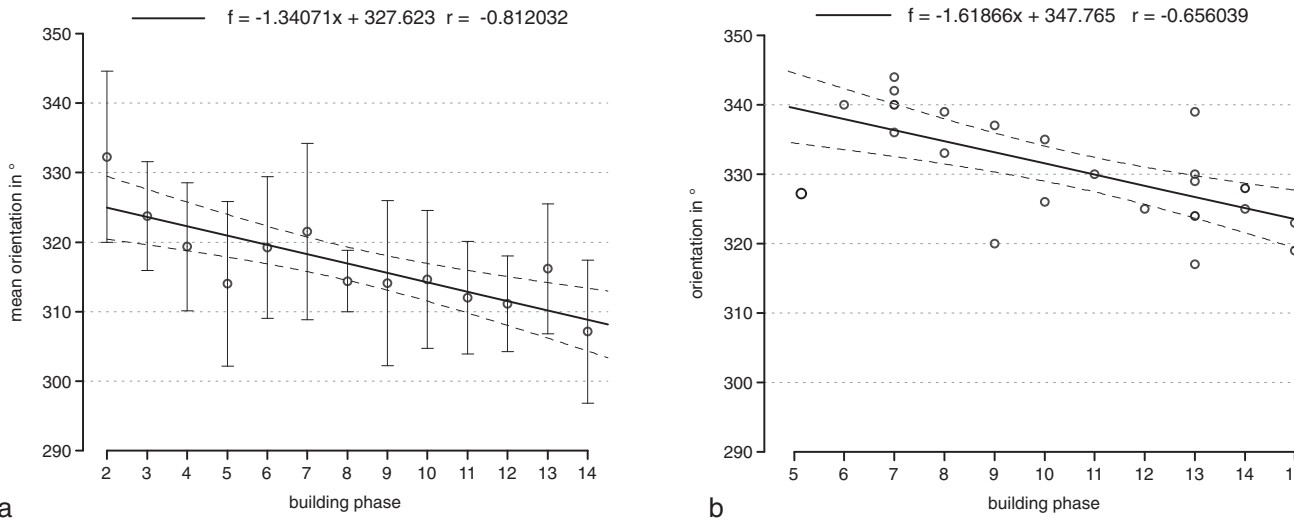

**Fig 7. Orientation and building phases of houses in the Rhineland.** a „Aldenhovener Platte"and „Hambacher Forst"(n = 128; mean orientation and standard deviation data after [31], 10 Abb. 10); b Erkelenz-Kückhoven (n = 26; data after [33]). Regression lines with 95%-confidence intervals.

325˚, later houses are clearly oriented more to the west (the regression line ending at 308˚). Across the 13 phases of an assumed length of 25 years each [32], this equals 5˚ over 100 years, 1˚ every 19 years, or 1.3˚ from phase to phase (see regression equation).

The same tendency is also well observable for the large settlement of Erkelenz-Kückhoven (Fig 7B). 26 houses are datable according to [33]. Grouped by phases, the average orientation shows a very high correlation with age. The oldest houses have a westward orientation of around 20˚ west from true North (= 340˚), the youngest houses deviate to about 35˚ west from North (= 325˚). During 11 phases, each lasting ca. 25 years (i.e. 275 years in total), the orientation of the houses thus shifted on average 15˚. This equates with a deviation of 5˚ over 100 years, 1˚ every 18 years, or 1.6˚ from phase to phase (see regression equation).

In eastern France, in Alsace, a change in house orientation was observed in several settlements [34–36]. The phasing of the houses is based on a thorough ceramic sequence that was used for Bayesian modeling of high quality ${}^{14}$C-dates [37]. In Bischoffsheim, houses which are attributed to the oldest phase have orientations between 283˚ and 335˚, with the majority oriented between 300˚ and 320˚. The houses of the middle phase show directions between 290˚ and 305˚. Finally, the latest houses face 290˚ ([36], 23f. with Fig 7).

Another local region where a change in orientation was observed is the area north of the Harz Mountains ([38], 172; 176). While the oldest houses are oriented north-south, the younger ones are directed northwest-southeast.

Taking the data from this sample of well-researched micro-regions together– Alsace, Merzbach valley, northern Harz, and southwest Slovakia– there can be little doubt that there is a temporal shift in house orientation to be found on a local scale across the LBK. In all cases this shift is counter-clockwise, whether from an original northeast more towards true north or from an original northwest towards west.

## Discussion

### House orientation as a measure of time

In our view, these findings can most plausibly be explained with as a slow, continuous process of a uni-directional change of orientation through time. An increasing number of house plans

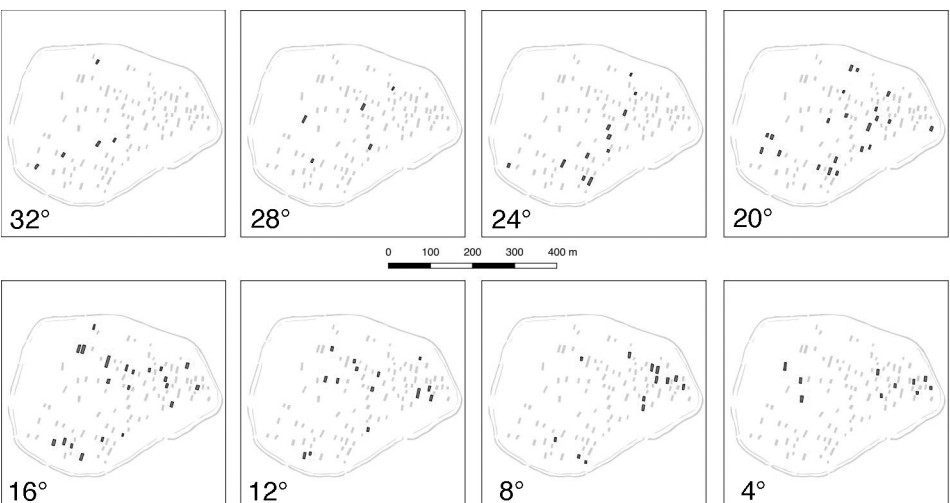

**Fig 8. Orientation of houses of the southwestern settlement of Vráble in steps of 4˚.**

over the course of the settlement histories leads to decreasing distances between neighboring houses. Apparently an increase in the overall variability in orientation accompanies this process. The duration of a settlement's occupation, the density of houses within it, and the variability of orientation of houses seem to be connected. The fact that higher density of buildings is linked to increasingly deviating orientations between immediate neighbors implies that houses were usually not built very close to older houses, otherwise we would expect a closer alignment between the older and younger houses.

The fact that houses with similar orientations are found at regular intervals of 75 and 150 m is in our view a clear argument for the existence of house wards ("Hofplatzmodell") as has been argued in the past [39]. With a distance of 75 m (taking the value of 150 m as the double of the regular distance; for the methodology see [40]) between contemporary houses, such a house ward would have encompassed roughly 0.5 ha. The buildings of one house ward were erected over time within this circumscribed area of 0.5 ha, resulting in the emergence of the described pattern as the buildings' show more and more changes in their orientation.

The regression analysis of orientation and age at Vráble reveals that orientation shifted on average by between 0.03˚ (southeastern settlement), 0.13˚ (southwestern settlement) and 0.18˚ (northern settlement) per year or between 3˚, 13˚ and 18˚ in 100 years. For the Vráble settlement as a whole the change amounts to 0.05˚ per year or 5˚ in 100 years.

The most reliable value can be attributed to the southwestern settlement as here, as already emphasized above, we have the best coverage. The fact that Bayesian $^{14}$C-modeling implies a broad co-existence of the three settlements [21] correlates with the observation of very similar means of the orientation of houses (see above). Therefore, we argue that the differences in the regression analyses of the three settlements are due to uneven sampling and do not reflect fundamental different principles in orientating houses.

If we interpret the differing orientation as a proxy for the individual dating of houses, the spatial development of the settlement can be reconstructed both in time intervals and as a steady process. Consequently, interesting conclusions regarding the history of the southwestern settlement can be drawn (Fig 8). Firstly, there is no clear spatial pattern visible concerning a possible direction of settlement development, say, from north to south or east to west. Already at the beginning of occupation, in the first phases, houses seem to cover almost the entire 14 ha area. This implies, secondly, that already at the very beginning of the settlement,

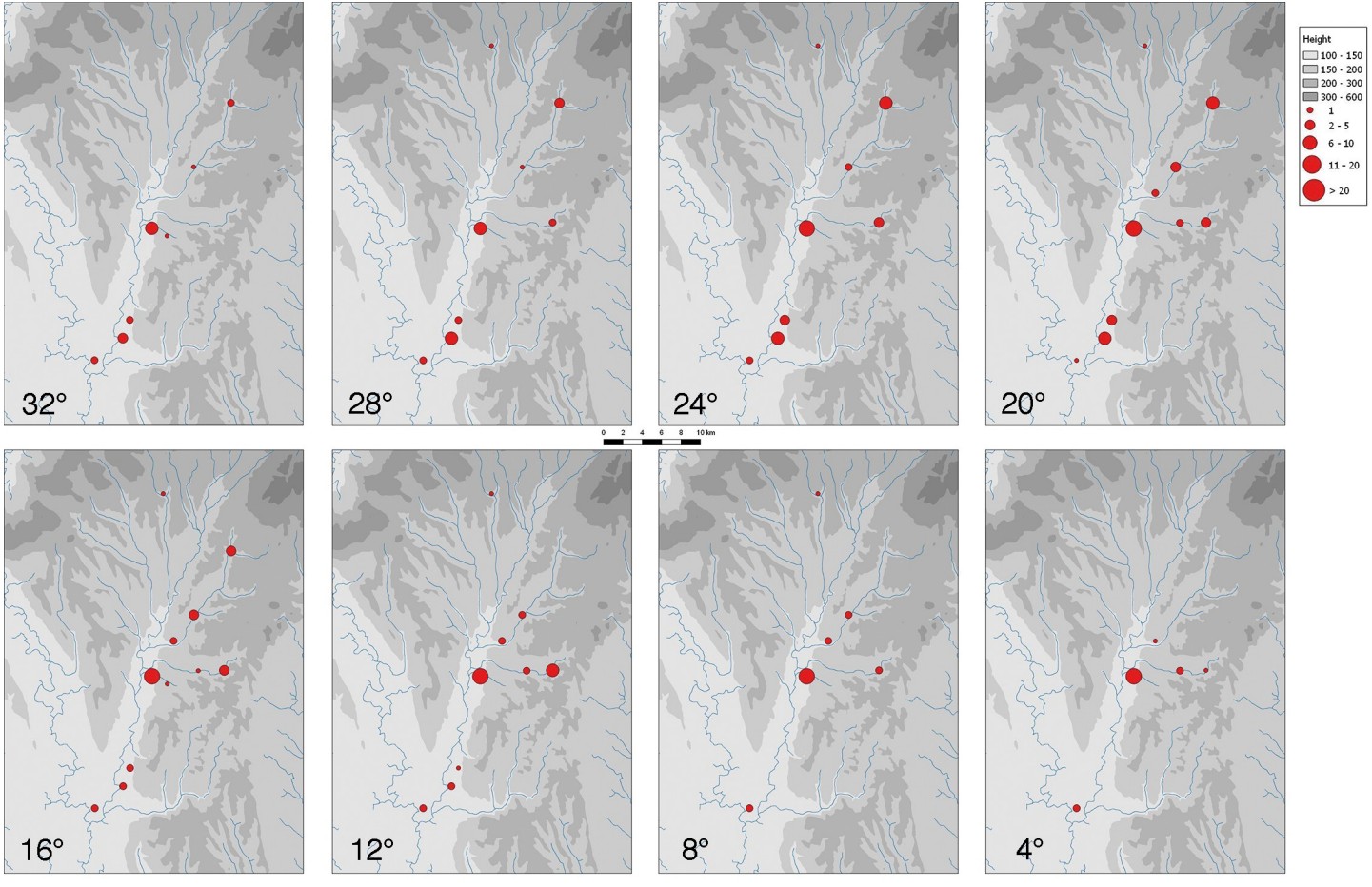

**Fig 9. Orientation of houses of the LBK-settlements of the Žitava-valley in steps of 4˚.**

the final size it would reach was implicitly or explicitly determined. This is not unlikely given that all three settlements in Vráble have the same size and shape. The only obvious exception is the northeastern-most part of the southwestern settlement where only houses with orientations less than or equal to 20˚ appear. Already judging by the settlement layout it was suspected that this appendix-like part of the settlement might be somewhat younger that the rest. The fact that the ditches, however, encircle even this appendix can be taken as a hint that the ditch system was erected only in the second half of the existence of Vráble, or even towards its end. Finally, after a modest beginning in the first two phases of occupation (c. 5275–5225 BCE), the settlement remains at a densely populated level with 10 houses or more, and with even more than 20 houses in the phases representing orientations of $\leq 20˚$ and $\leq 16$ (c. 5150–5100 BCE).

## Change in orientation in the Žitava valley

Mapping the 4˚-interval (roughly equalling phases of 40 years and thus the length of the occupation of individual houses [see above] between 5275–4950 BCE) for all magnetically prospected settlements of the Žitava-valley, we can model a chronological sequence in which a process of settlement concentration within this micro-region emerges (Figs 9 and 10).

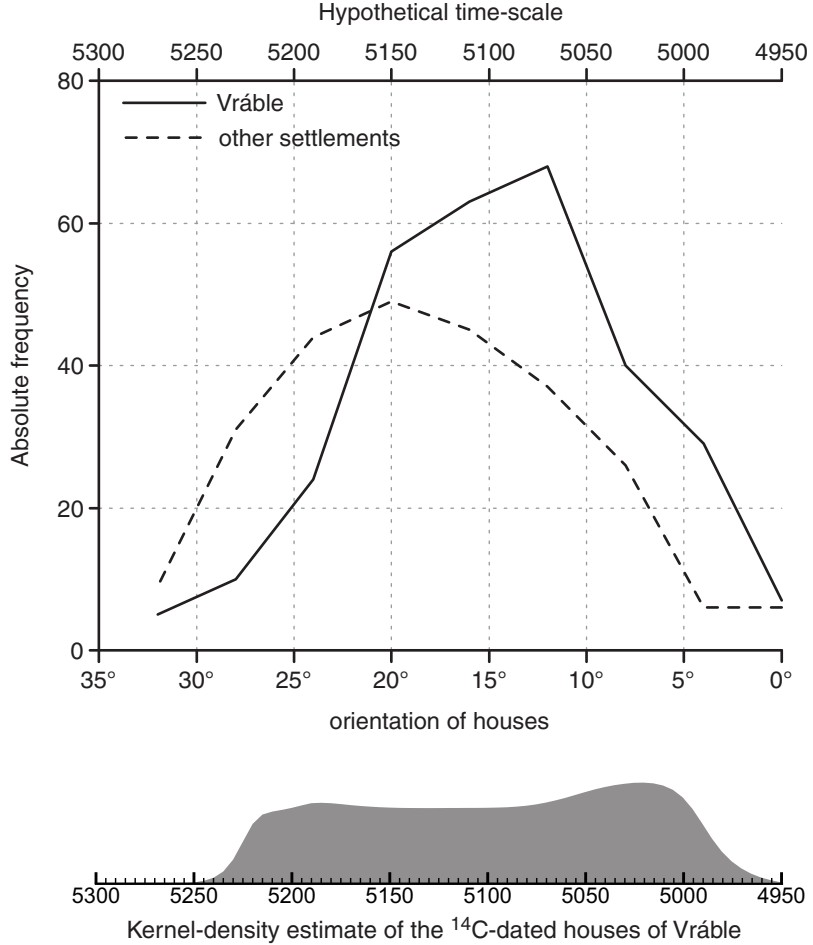

**Fig 10. Hypothetical settlement development in the Žitava-valley based on house orientations and compared to KDE-modeled $^{14}$C-dates (function „KDE_plot"of OxCal 4.3).**

While the settlement of Vráble is–apart from Úĺĺany–the only settlement that is present on all maps, it seems to have reached a remarkable position in terms of size only after approximately 100 years (orientation $</$ = 24˚, c. 5175 BCE). However, from then onwards it is the only settlement with more than 30 contemporary houses. This is especially clear in the last phases (orientation $</$ = 8˚, ca. 5025–4950 BCE) where in most other sites of the Žitava valley LBK-settlement seems to have ceased completely. The interpretation of the magnetic prospections thus underlines the special status of Vráble probably as a central site, comparable for example to Langweiler 8 in the well-researched Merzbachtal [41]. However, it seems to have gained this prominent position only towards the end of the local LBK and at the expense of other settlements' existence. In consequence, we are able to detect an agglomeration process of the local population.

The maps imply that the Žitava valley was already fully settled when the settlements represented by magnetic plans were founded. Assuming a steady change in orientation, the houses depicted in the magnetic imagery can be hypothetically pin-pointed mainly to a duration of 300 years and the time-range 5250–4950 calBC. According to $^{14}$C-dates, the change from the so-called younger LBK (Notenkopfkeramik) to the Želiezovce-group, the youngest variant of local LBK, happened around 5200 calBC [21]. Therefore, some of the identified houses should

belong to the Notenkopfkeramik phase. This is in accordance with ceramic surface finds which imply that older LBK settlements already reached even remote parts of the Upper Žitava valley and its tributaries ([42], 205 map 4 no. 36.59).

## Explanation: Pseudoneglect

The supra-regional differences and tendencies in orientation across the LBK world have been the topic of intensive discussions in the past and have been explained with a wide variety of theories (climate/dominating wind directions: [43]; home of the ancestors: [44]; exposure to the sun: [10]). Astonishingly, the variability of house orientation within individual settlements was never investigated in more detail. As mentioned above, the suggestion by Sangmeister that variability in orientation could be due to chronological differences was never seriously considered or even explicitly refuted (see above). As we have shown, there is in fact a directed shift in orientation over time. However, this shift is so small that it seems very unlikely that the inhabitants of the houses and settlements would realistically have been aware of it. When we consider the maximum life-span of an LBK house of 25–50 years (although there are scholars who maintain that these houses can last up to 100 years) [45]), the average difference of house orientations in adjacent phases would be 3–5˚. This is barely perceptible when looking at a ground plan from above and would be virtually impossible to detect from the ground. It seems very unlikely that contemporary inhabitants of the settlements would have any chance to notice the difference. As a corollary, we highly suspect that the change in orientation happened unconsciously and therefore not deliberately.

We do not see any celestial body that could be held accountable for this phenomenon, as the starting orientations in the different areas of the LBK were very different. Moreover, similar processes were also observed in completely different archaeological contexts, e. g. in Iron Age houses ([46], 14–17) or Iron Age burials ([47]; here–perhaps erroneously–explained with the precession of the stars in the sky, though). Therefore, it seems most probable that a certain inclination in human perception lead to such a shift in the long-term.

Pseudoneglect could be the source of such an inclination. This term refers to the fact that neurologically healthy individuals typically privilege the left side of space (they „neglect"the right side) and hence bisect a horizontal line to the left of its veridical center (see [48] for an extensive meta-study). Accepting this explanation, it can be argued that humans faced with the task of erecting a new house at a certain distance from other houses and with the firm determination to align it exactly with these older houses, would, in the absence of devices for exact measurements, most certainly misperceive the intended orientation. Pseudoneglect would lead to a non-random deviation of the aimed-at orientation; the accomplished orientation would on average lie somewhat to the left of the true orientation which in the long run would translate into a counter-clockwise rotation of houses.

Differing from the original formulation by Sangmeister [11], we do not claim that the change in orientation of houses is due to a complete abandonment and resettling of a certain settlement with a new orientation, but that the change happened on a continuous and constant basis, where individual house wards rebuilt their houses in a slightly and barely perceptible progressively counter-clockwise altered orientation due to pseudoneglect.

## Caveats: Close proximity of houses

For the settlement Langweiler 8, it was considered that not contemporaneity, but the spatial proximity of houses was the primary reason for a similar orientation ([30], 925). However, Mattheußer ([31], 16) could show that the orientation of a specific house is not tied to that of the Nearest Neighbor (see also above). On the other hand, in those cases where groups of

similar alignments exist, she speculates that sometimes new houses were built so close to older houses that they were forced to follow their orientations, causing alignment. As a result, the temporal differences would be blurred by spatial proximity.

Exactly such a case was encountered in Vráble in the 2016 excavation area. There, the magnetic picture seemed to imply three houses standing very close to each other with the same orientation (houses 131, 132, 133). As working hypothesis this situation was interpreted as a quarter and as perhaps a particularly well preserved example of a row-like arrangement of houses. However, during excavation it turned out that the houses were not contemporaneous, and that parts of their long pits stratigraphically intersect [23]. Not surprisingly, the raw and modeled [14]C-dates mirror this time-depth [21]. Therefore, some of the houses from the 2016 excavation area are not in concordance with the orientation = dating hypothesis (Fig 6). However, such cases should be conceived as cautionary tales rather than grounds on which to abandon the hypothesis altogether.

## Consequences

Our findings have fundamental consequences for the interpretation of large scale settlement plans. Even when accepting all possible caveats–for example the possibility of adjacent but not contemporaneous houses acting as guides and thus obscuring the temporal effect–the possibilities of investigating settlement plans without extensive excavations are intriguing. In the overwhelming majority of LBK-settlements, the excavation of all houses is neither possible nor desirable because of preservation considerations or financial constraints. The concentration on differences in orientation offers the possibility to still make substantial statements about the development of individual settlements or even micro-regions. In this regard, our conclusions in relation to the settlement history of the LBK Žitava valley–the deliberate planning of settlements from the beginning of occupation, contemporaneity of 10 households on average in Vráble, a demographic agglomeration process–barely skim the realm of what is possible with intense geophysical surveys and targeted dating of features.

While our discussion was mainly focused on the interpretation of magnetic surveys, the possibility of interpretation is by no means limited to it. The availability of large-scale excavation plans offers the opportunity to extend the approach to this medium as has already demonstrated above with the example of Erkelenz-Kückhoven.

## Conclusion

We have shown that the shift in orientation of longhouses can be consistently documented at multiple archaeological sites of the Early Neolithic and that this shift has a chronological dimension. As explanation we propose that this is due to ‚pseudoneglect‘ which is well documented in a multitude of laboratory studies as a neurobiological aspect of human perception. We hope that our findings will be put to the test in other contexts, archaeological as well as modern ones. Apart from controlled studies, we would expect to find similar developments in all situations where there is a continuity of built space and where the building and rebuilding process of houses is not governed by strict rules or overarching authorities.

## Acknowledgments

We thank Sarah Martini for English proofreading of the manuscript and two anonymous reviewers for their helpful suggestions.

## Author Contributions

**Conceptualization:** Nils Müller-Scheeßel, Martin Furholt.

**Data curation:** Nils Müller-Scheeßel, Wiebke Mainusch, Knut Rassmann, Erica Corradini.

**Formal analysis:** Nils Müller-Scheeßel.

**Funding acquisition:** Johannes Müller, Wolfgang Rabbel, Martin Furholt.

**Investigation:** Nils Müller-Scheeßel, Ivan Cheben, Wiebke Mainusch, Knut Rassmann, Erica Corradini, Martin Furholt.

**Methodology:** Nils Müller-Scheeßel.

**Project administration:** Ivan Cheben, Wolfgang Rabbel, Martin Furholt.

**Resources:** Knut Rassmann.

**Supervision:** Johannes Müller, Martin Furholt.

**Visualization:** Nils Müller-Scheeßel.

**Writing – original draft:** Nils Müller-Scheeßel, Johannes Müller, Martin Furholt.

**Writing – review & editing:** Nils Müller-Scheeßel, Johannes Müller, Wolfgang Rabbel, Martin Furholt.

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
