## [Decision Letter · Decision Letter 0]

9 Oct 2019

PONE-D-19-18502

A new key method for settlement archaeology: dating Linearbandkeramik houses by their orientation

PLOS ONE

Dear Dr. Müller-Scheeßel,

Thank you for submitting your manuscript to PLOS ONE. After careful consideration, we feel that it has merit but does not fully meet PLOS ONE’s publication criteria as it currently stands. Therefore, we invite you to submit a revised version of the manuscript that addresses the points raised during the review process.

All comments need to be addressed.

We would appreciate receiving your revised manuscript by Nov 23 2019 11:59PM. To enhance the reproducibility of your results, we recommend that if applicable you deposit your laboratory protocols in protocols.io, where a protocol can be assigned its own identifier (DOI) such that it can be cited independently in the future. For instructions see: http://journals.plos.org/plosone/s/submission-guidelines#loc-laboratory-protocols

We look forward to receiving your revised manuscript.

Kind regards,

Peter F. Biehl, PhD

Academic Editor

PLOS ONE

Journal Requirements:

1. We note that you have stated that you will provide repository information for your data at acceptance. Should your manuscript be accepted for publication, we will hold it until you provide the relevant accession numbers or DOIs necessary to access your data. If you wish to make changes to your Data Availability statement, please describe these changes in your cover letter and we will update your Data Availability statement to reflect the information you provide.

Academic editor comments;

Your manuscript has now been seen by two referees, whose comments are appended below. You will see from these comments that while the referees find your work of potential interest, they have raised substantial concerns that must be addressed. In light of these comments, we cannot accept the manuscript for publication, but would be interested in considering a revised version that addresses these serious concerns.

We hope you will find the referees' comments useful as you decide how to proceed. Should presentation of further data and analysis allow you to address these criticisms, we would be happy to look at a substantially revised manuscript. However, please bear in mind that we will be reluctant to approach the referees again in the absence of major revisions.

Reviewers' comments:

Reviewer's Responses to Questions

**Comments to the Author**

1. Is the manuscript technically sound, and do the data support the conclusions?

Reviewer #1: Yes

Reviewer #2: Partly

2. Has the statistical analysis been performed appropriately and rigorously? 

Reviewer #1: Yes

Reviewer #2: Yes

3. Have the authors made all data underlying the findings in their manuscript fully available?

Reviewer #1: Yes

Reviewer #2: Yes

4. Is the manuscript presented in an intelligible fashion and written in standard English?

Reviewer #1: Yes

Reviewer #2: Yes

5. Review Comments to the Author

Reviewer #1: This is an interesting paper. I would make several suggestions for improvement. First, the title is overly specific. Remember that PLoS ONe is not an archaeological journal. It has a broad scientific readership. Thus I would avoid using archaeological terms like Linearbandkeramik in the title. Second, I suggest that an introductory paragraph be added to show the reader the importance of this study and why the reader should be interested in it. Explain the nature of settlements with longhouses and situate them chronologically. Finally, the section labeled "Conclusion" should clearly state the conclusions rather than continue the discussion. The main conclusion is that the shift in orientation of longhouses can be consistently documented at multiple sites and that is has a chronological dimension. The proposed explanation is the psychological aspect of human perception. Obviously, this is a hypothesis rather than a conclusion, but you could point to some ways in which this hypothesis can be tested.

Reviewer #2: The approach is ok, if not - to a certain extend - without humerous aspects. However, personally, I am not convinced that this hypothesis really explains the long-known phenomenon of the gradual westward - or eastward in eastern Europe - shift of Early Neolithic long houses accross Europe. It was developed at this fascinating and absolutely outstanding site, where it makes some sense, if the site is considered in isolation. If, however, the entire data base is considered, there are numerous arguments against it. First and foremote is the regional variation considerable, also according to local topography. Then, at some sites we can observe shifts in west-ward orientation which is a counterargument against gradual visual misconception.

Lastly, the entrie argument of time-sinsitivity of house orientation is not new. I therefore strongly suggest to rephrase the argument in the sense that another hypothesis os offered, which has some power for the site of Vráble, but apparently has its weak points even for that very same site, as the authors themselves hesitantly admit (line 381-390).

This also requires a reformulation of the title, maybe :"A new hypothesis on the orientation of Early Neolithic long houses".

6. PLOS authors have the option to publish the peer review history of their article (what does this mean?). If published, this will include your full peer review and any attached files.

Reviewer #1: No

Reviewer #2: No

---

## [Author Response · Author response to Decision Letter 0]

11 Nov 2019

Title (l. 1-2)

Both reviewers were unhappy with the title. On the one hand, it was deemed too specific (reviewer 1 + 2), on the other that it should be rephrased (reviewer 2). We followed both suggestions and rephrased the title completely.

Introduction (l. 32-75)

Reviewer 1 made the valuable suggestion to state in an additional introductory paragraph why the reader should be interested in the paper and to situate the study in time and space. We have added such a paragraph.

Conclusion (l. 393-414)

Reviewer 1 demanded that the conclusion should be really a conclusion and not an extension of the discussion, and that it should name additional avenues for testing the hypothesis. We have changed the conclusion accordingly.

Hypothesis „Change in orientation“

Reviewer 2 criticizes that „the entire argument of time-sensitivity of house orientation is not new“. We do not claim to have invented this hypothesis but on the contrary discuss the early suggestions by E. Sangmeister. But, in contrast to Sangmeister, we are able to present data to underpin the hypothesis.

Reviewer 2 states that s/he is not convinced by our line of argument at all. We would like to point out that obviously s/he misconceived some of our arguments. First and foremost, we do not deal with „the long-known phenomenon of the gradual westward - or eastward in eastern Europe - shift of Early Neolithic long houses accross Europe.“ We only look at change in orientation PER SITE (or small scale region, in the case of the Merzbachtal).

We completely agree that „the regional variation [is] considerable“ but in our view the question why this is the case is a completely different topic. Local studies explicating „jumps“ in orientation concern, for example, the surrounding of Dresden (Link Th [2011] Böhmische Dörfer? Zur Stellung der Dresdener Elbtalweitung zwischen sächsischer und böhmischer Bandkeramik. In Th. Doppler, B. Ramminger, D. Schimmelpfennig (eds.) Grenzen und Grenzräume? Kerpen-Loogh: 11-24) or Southern Bavaria (Pechtl J [2010] Anmerkungen zum Kenntnisstand linienbandkeramischer Hausarchitektur im südöstlichen Bayern und zum Potenzial ihrer typologischen Auswertungen. Fines Transire 19: 35-51), and in our view such „jumps“ are highly significant, but beyond the scope of the present paper.

It has been claimed again and again that the variation in orientation follows „local topography“ (e. g., because of prevailing currents). However, where this suggestion was put to the test (Mattheußer E (1991) Die geographische Ausrichtung bandkeramischer Häuser. In: Studien zur Siedlungsarchäologie 1. Bonn: Habelt. pp. 1-49), it turned out to be not valid. Anyway, even if true, it would not explain variation per site. 

We would be keen to hear of the „some sites [where] we can observe shifts in west-ward [= clockwise?] orientation which is a counterargument against gradual visual misconception.“ We actually looked at a much larger number of sites than presented in the paper but only found the same tendency of counter-clockwise shifts. Bylany may be the only possible exception when it comes to major excavated sites. Unfortunately, the available data is presented in such a condensed manner (Pamatky archeologicke 77, 1986, 397 Table 49) that it is difficult to draw definite conclusions.

The point is, and this is clearly stated in text, that the pseudoneglect explanation only is relevant where there are no other reasons (topography, agglomeration of houses, circular arrangement of houses). There might also be other factors we have not accounted for, or that we archaeologically cannot account for. However, the evidence of counter-clockwise shifts is strong, and it is clearly not to be dismissed even if there were singular occurences of clockwise ones.

To account for the critique of reviewer 2, we added explaining sentences at the beginning, in the middle and at the end.

---

## [Decision Letter · Decision Letter 1]

20 Nov 2019

A new approach to the temporal significance of house orientations in European Early Neolithic settlements

PONE-D-19-18502R1

Dear Dr. Müller-Scheeßel,

We are pleased to inform you that your manuscript has been judged scientifically suitable for publication and will be formally accepted for publication once it complies with all outstanding technical requirements.

With kind regards,

Peter F. Biehl, PhD

Academic Editor

PLOS ONE

Additional Editor Comments (optional):

Reviewers' comments:

Reviewer's Responses to Questions

**Comments to the Author**

1. If the authors have adequately addressed your comments raised in a previous round of review and you feel that this manuscript is now acceptable for publication, you may indicate that here to bypass the “Comments to the Author” section, enter your conflict of interest statement in the “Confidential to Editor” section, and submit your "Accept" recommendation.

Reviewer #1: All comments have been addressed

Reviewer #2: All comments have been addressed

2. Is the manuscript technically sound, and do the data support the conclusions?

Reviewer #1: (No Response)

Reviewer #2: Yes

3. Has the statistical analysis been performed appropriately and rigorously? 

Reviewer #1: (No Response)

Reviewer #2: Yes

4. Have the authors made all data underlying the findings in their manuscript fully available?

Reviewer #1: (No Response)

Reviewer #2: Yes

5. Is the manuscript presented in an intelligible fashion and written in standard English?

Reviewer #1: (No Response)

Reviewer #2: Yes

6. Review Comments to the Author

Reviewer #1: I am grateful that the authors took my earlier comments and those of Reviewer 2 into consideration and responded to them thoughtfully. This is an important complex of sites that poses methodological challenges in their interpretation, and this paper presents an approach that can be tried at other sites and the results can be compared.

Reviewer #2: (No Response)

7. PLOS authors have the option to publish the peer review history of their article (what does this mean?). If published, this will include your full peer review and any attached files.

Reviewer #1: No

Reviewer #2: No

---

## [Editor Report · Acceptance letter]

17 Dec 2019

PONE-D-19-18502R1 

A new approach to the temporal significance of house orientations in European Early Neolithic settlements 

Dear Dr. Müller-Scheeßel:

I am pleased to inform you that your manuscript has been deemed suitable for publication in PLOS ONE. Congratulations! Your manuscript is now with our production department. 

With kind regards,

on behalf of

Dr. Peter F. Biehl 

Academic Editor

PLOS ONE